# Application of Micro-Computed Tomography for the Estimation of the Post-Mortem Interval of Human Skeletal Remains

**DOI:** 10.3390/biology11081105

**Published:** 2022-07-25

**Authors:** Verena-Maria Schmidt, Philipp Zelger, Claudia Woess, Anton K. Pallua, Rohit Arora, Gerald Degenhart, Andrea Brunner, Bettina Zelger, Michael Schirmer, Walter Rabl, Johannes D. Pallua

**Affiliations:** 1Institute of Legal Medicine, Medical University of Innsbruck, Muellerstraße 44, 6020 Innsbruck, Austria; verena-maria.schmidt@i-med.ac.at (V.-M.S.); claudia.woess@i-med.ac.at (C.W.); walter.rabl@i-med.ac.at (W.R.); 2University Clinic for Hearing, Voice and Speech Disorders, Medical University of Innsbruck, Anichstrasse 35, 6020 Innsbruck, Austria; philipp.zelger@tirol-kliniken.at; 3Former Institute for Computed Tomography-Neuro CT, Medical University of Innsbruck, Anichstraße 35, 6020 Innsbruck, Austria; anton.k.pallua@cnh.at; 4University Hospital for Orthopaedics and Traumatology, Medical University of Innsbruck, Anichstraße 35, 6020 Innsbruck, Austria; rohit.arora@tirol-kliniken.at; 5Department of Radiology, Medical University of Innsbruck, Anichstraße 35, 6020 Innsbruck, Austria; gerald.degenhart@i-med.ac.at; 6Institute of Pathology, Neuropathology, Molecular Pathology, Medical University of Innsbruck, Muellerstrasse 44, 6020 Innsbruck, Austria; andrea.brunner@i-med.ac.at (A.B.); bettina.zelger@i-med.ac.at (B.Z.); 7Department of Internal Medicine, Clinic II, Medical University of Innsbruck, Anichstrasse 35, 6020 Innsbruck, Austria; schirmer.michael@icloud.com

**Keywords:** post-mortem interval, micro-CT, machine learning

## Abstract

**Simple Summary:**

With a short sample-preparation time, micro-computer tomography provides a non-destructive method to estimate the post-mortem interval. With a deep learning approach for post-mortem interval estimation (ranging from one day to 2000 years) in bones, the estimation can be approximated with high precision.

**Abstract:**

It is challenging to estimate the post-mortem interval (PMI) of skeletal remains within a forensic context. As a result of their interactions with the environment, bones undergo several chemical and physical changes after death. So far, multiple methods have been used to follow up on post-mortem changes. There is, however, no definitive way to estimate the PMI of skeletal remains. This research aimed to propose a methodology capable of estimating the PMI using micro-computed tomography measurements of 104 human skeletal remains with PMIs between one day and 2000 years. The present study indicates that micro-computed tomography could be considered an objective and precise method of PMI evaluation in forensic medicine. The measured parameters show a significant difference regarding the PMI for Cort Porosity *p* < 0.001, BV/TV *p* > 0.001, Mean1 *p* > 0.001 and Mean2 *p* > 0.005. Using a machine learning approach, the neural network showed an accuracy of 99% for distinguishing between samples with a PMI of less than 100 years and archaeological samples.

## 1. Introduction

Estimating the post-mortem interval (PMI) is challenging in forensic medicine [1,2,3]. It is crucial to calculate the PMI when detecting human remains. It is well known that bones undergo various chemical and physical processes after death due to their interaction with the environment in which they are located. Several methods have been used to track these post-mortem changes to estimate the elapsed time since a person’s death, but they have serious drawbacks in terms of reliability and accuracy [4,5]. Various techniques and methods have been employed to estimate the PMI with the highest accuracy. Methods include examination of the external physical appearance; histopathological surveys [6,7,8]; reaction with a mineral acid, reaction with benzidine, nitrogen loss [9]; molecular biology [10,11,12,13,14,15]; metabolomics [16]; high performance liquid chromatography-tandem mass spectrometry [17]; UV-Vis spectroscopic methods [18,19,20,21,22]; radioisotope measurements [23,24,25,26]; luminol chemiluminescent reaction [24,27,28,29,30,31]; X-ray diffraction [32,33,34]; spectroscopic technology [21,34,35,36,37,38,39,40,41,42,43,44,45,46,47]; postmortem computed tomography (CT) [48], micro-CT [2,34]; visible and thermal 3D imaging [49]; and entomological methods (succession model, carrion insect development) [50]. No matter what technique is used, decomposition of tissue with time makes estimating PMI difficult [51]. Despite these favourable factors, the precise estimation of PMI is not possible with the current approaches. This makes micro-computed tomography of the bone extremely beneficial in PMI estimation since it provides the most accurate information about the hard tissues in the body [2,34,52]. The significant advantage of the micro-CT is that it allows the evaluation of small specimens and small alive animals due to their capacity or small scanning chambers.

Furthermore, getting qualitative and quantitative results with high-resolution images and small samples provided several applications in the in vivo and in vitro imaging of bones [2,52]. The micro-CT technique is similar to CT in terms of its physical and technical basis. In preclinical research, they are essentially miniaturized versions of the volume- or cone-beam CT scanner used for non-invasive, three-dimensional studies of bones, teeth, and small animals. Micro-CT offers several advantages over clinical CT, including a significantly higher spatial resolution and improved anatomical structure visualization [53,54,55]. The latest equipment allows for in vivo measurements down to a spatial resolution of 10 µm [56,57]. The present study hypothesized that bone density could be used to evaluate mineral density in different post-mortem timelines as an indicator of PMI. To achieve this goal, micro-CT, one of the most novel and accurate methods for quantitative imaging, is used.

## 2. Materials and Methods

### 2.1. Sample Collection and Ethical Considerations

Recent forensic bone samples (*n* = 99) were routinely collected for molecular genetic identification purposes during an autopsy at the University Institute of Forensic Medicine, and archaeological bone samples from medieval times were collected from European excavation sites (*n* = 5). The bone samples with 0–2 weeks PMI (class 1, *n* = 32), 2 weeks–6 months PMI (class 2, *n* = 46), 6 months–1 year PMI (class 3, *n* = 11), 1 year–10 years PMI (class 4, *n* = 10), and > 100 years PMI (class 5, *n* = 5) were obtained from 16 female and 88 male human remains. The classification of PMI was based on investigations by the police and forensic needs before micro-CT. In the case of the uncertain conventional estimation of PMIs, the average result was used for classification. The diaphysis of the femur of forensic and archaeological bone samples was used for analyses. Using a hand saw, one half transversal section was cut from each bone with a thickness of about 7 mm. The sectional planes were cleaned from periost and bone marrow and dried a few days at room temperature. NIR spectrometry was applied for this study before additional forensic analyses.

The study was conducted according to the ICH-GCP guidelines and the declaration of Helsinki. Ethical approval was obtained from the local ethics commission (EK: 1357/2021).

### 2.2. Micro-CT

Micro-CT experiments were performed using a vivaCT 40 (Scanco Medical AG, Brüttisellen, Switzerland). The scan settings were 1000 projections with 2048 samples, resulting in a 15 µm isotropic resolution using a 30.72 mm field of view. The used tube settings were 70 kV voltage, 114 µA current, and an integration time of 200 ms per projection. The acquired images have a 2048 × 2048 voxels matrix and a grayscale depth of 16 bit. The length of the image stack is individually dependent on the size of the collected femora sample. While most samples could be positioned axially in the sample holder, some had to be longitudinally scanned due to size restrictions. The image reconstruction using a cone beam convoluted back-projection and post-processing was performed using the system workstation of the micro-CT. The system workstation is running on open VMS (©Hewlet Packard, Palo Alto, Santa Clara, CA, USA) combined with IPL (Image processing language, Image Processing Language, Scanco Medical AG, Brüttisellen, Switzerland). The post-processing consisted of two independent steps. The IPL integrated alignment algorithm was used for the longitudinal scanned samples to rotate the image space that the samples were represented axially.

### 2.3. Segmentation and Quantitative Analysis

For the evaluation, four separate areas were analyzed. The first area was the whole cortical bone (Figure 1A). The following areas were 3200 pixels wide, 200-pixel long cylinders aligned centrally in the cortical bone around the analyzed sample (Figure 1B). The segmentation of the bone used the standard parameters for the Gauss threshold segmentation (threshold: 220, gauss sigma: 0.8, gauss support: 1). The bone parameters evaluated are summarized in Table 1.

### 2.4. Statistical Analysis and Machine Learning

ANOVA and a machine learning approach were used to analyze the time-dependent data. The ANOVA was performed on time-dependent parameter sets obtained from the micro-CT scans. Additionally, ANOVA analysis was performed to check whether the data obtained from the whole cortical bone or cylinders aligned centrally in the cortical bone around the analyzed sample showed a statistically significant difference. The Shapiro–Wilk test for normality was used to check whether the data met the requirements for the ANOVA analysis. The micro-CT data have been further analyzed using a machine learning approach. A small, fully connected neural network (3 layers, 10, 25, 10 neurons per layer) was implemented. The micro-CT data was used to train and evaluate the predictive performance of the neural network. The output of the neural network, consisting of 5 binary neurons (sigmoid activation function), corresponds to the belonging to the respective age class. The neural network was trained using the leave-one-out cross-validation (LOOCV). With the LOOCV method, one data point is left out in the training process. This data point is then used to evaluate the predictive abilities of the neural network. This step (a new training from scratch) is repeated until every data point has been left out and assessed once. This technique helps to deal with the limitations of small data sets.

To exclude a possible bias by premortem loss of bone, the BV/TV ratio and the Cort porosity were then age-adjusted (where the age at death was available) according to a Canadian population-based HR-pQCT study [58]. The data of [58] demonstrated a linear or quadratic behaviour between the ages of 20 and 60. That made it numerically favourable to choose an age in this range. From this range, the median age was taken. However, the choice for the base age is not expected to change the outcome of the analysis since all data were normalized to that age. Therefore, the values have been normalized to the 40-year age value, and data compared between the different age classes as described above.

## 3. Results

The data obtained by micro-CT provide a high dimension of information, demonstrating different structures or mechanisms linked to the PMI. Optimizing the analyzing strategy for micro-CT experiments is essential to get high-quality results. All samples were analyzed using the whole cortical bone for morphological analyses, and cylinders aligned centrally in the cortical bone around the analyzed sample. Results are illustrated in Table 2, Figure 2 for the whole cortical bone, Figure 3 for cylinders aligned centrally in the cortical bone, and Figure 4 boxplots of selected values from all the extraction locations. Comparison between the two strategies indicated negligible differences, concluding that using the whole cortical bone for analysis is sufficient for PMI estimation (see Table 2 and Figure 4). Therefore, it was possible to demonstrate that the internal and external decomposition processes hardly influence the evaluation strategy and micro-CT measurements. The 3-D surface renders images with local thickness analysis (as proposed by [59]), and separation from one representative sample for each analysis strategy is presented in Figure 2 for the whole sample and Figure 3 for cylinders aligned centrally. The microarchitecture was assessed using the standard trabecular segmentation and thickness values with a separation model [60,61,62,63,64,65]. Class 5 displays a higher cortical porosity (Cort Porosity) than the other classes, defined as trabecular separation and a decreased bone volume density (= bone volume over the total volume, BV/TV) (see Table 2).

Taken together, bone density and pore structure decrease over time (Table 2, Figure 2 and Figure 3). After normalizing the BV/TV-ratio and the Cort porosity to exclude a possible bias by bone loss before the death, data only changed by less than one standard deviation (data not shown).

### 3.1. Statistical Analysis and Classification of Post-Mortem Interval

The data of all measured samples using the whole bone information are compared using ANOVA. Figure 5 displays the boxplots of the corresponding data sets. The results show significant p values for cortical porosity (Cort Prososity), bone volume density (= bone volume over total volume, BV/TV), apparent density (Mean1) and material density (Mean2). A clear reduction in bone volume density (*p* < 0.001), apparent density (*p* < 0.001), and material density (*p* < 0.005) can be observed over time. The cortical porosity represented an increase in this value (*p* < 0.001).

### 3.2. Deep Learning-Based Classification

Figure 6A shows the classification result as a confusion matrix. The diagonal elements of such a confusion matrix represent the percentage of correctly classified elements. The bright diagonal elements of the confusion matrix represent the percentage of correctly classified elements, with an accuracy ranging between 99% for the archeological samples (class 5) and 75% for samples with a PMI between 0 and 2 weeks (class 1). Hence, 13 elements were classified with a PMI of “some weeks”, which allows an assignment to two classes. The result also indicates a cross-talk between neighboring classes, especially between the first two. This cross-talk is most likely due to the classes’ choice or proximity.

The same approach of deep-learning-based classification was then used for the premortem age-adjusted data. Results are comparable, as shown in Figure 6B.

## 4. Discussion

There is a need for forensic anthropologists and pathologists to continue investigating precise methods for PMI estimation of skeletal remains [56]. The determination of the PMI is based on an assessment of the morphological structures of the bones and an examination of the clothing and personal items found on the corpse. Furthermore, the police routinely provide information on the time and place of the discovery as well as the respective environmental conditions.

This study aimed to evaluate the suitability of micro-CT as a non-destructive method for distinguishing between forensic bone material with different PMIs and also archaeological bone material. Therefore, micro-CT measurements and a deep learning approach to correlate the PMI with the detected properties were used. The main focus of this study was the training of an ANN that allows for estimating the PMI of human bones with micro-CT to help authorities to assess if the found bone is from forensic interest or not. Depending on the criminal act committed, the statute of limitation varies or does not exist in the case of murder.

After optimizing the analyzing location (there was no difference between the extern and intern of the bones), we could show that the Cort Porosity increases with time and the bone density decreases over the years. The measured values reveal statistical significant differences between all the age classes.

As shown in a confusion matrix (Figure 6A), ANN classification results clearly distinguish between forensic and archaeological samples, but also showed differences within the forensic bone sample classes.

The result shows a perfect classification for the fifth class (archaeological samples) and the lowest accuracy (75%) for the first class. The age uncertainty of the first-class results in a blurring of the class boundary between the first and second class, e.g., a bone with a determined age of 14 (±1) days puts the dataset in both the first and second classes. The same effect, but not to the same magnitude, is present in all classes besides the fifth class. The small time window of the first class (0–14 days) compared to the age uncertainty in this range (0–7 days) amplifies this effect in the first class. The same problem can be found with the early samples of the second class. Some samples are classified with “some weeks”, which, within the error, allows an assignment to both classes. The same problem can be observed between the other classes besides the fifth class.

There are some limitations of this study. First, there was a high uncertainty in the PMI, as discussed above, despite the relatively high sample size. PMI was classified based on police investigations and forensic needs. The average PMI result was used for classification when conventional estimation was uncertain. The classification results might be explained by the differences in the chemical composition of bones. Differences in degradation and environmental effects might also explain the observed differences between the bones with different PMI. The classification results suggest that differences in the chemical composition of bones are responsible for the observed differences. Unfortunately, all information was not available in the set of samples analyzed. Samples were included with often undefined PMI descriptions, different find spots (e.g., forest, flat, buried, water), and thermal alterations (plane crash = 3 and apartment fire = 1).

Second, the premortem bone loss may lead to wrong results. Therefore, the data analysis was also performed with age-adjusted Cort Porosity and BV/TV values. The results did not change the classification accuracy so this limitation can be neglected in this setting.

The non-invasive nature of this analysis ensures the sample’s integrity before further analyses. Traditional methodologies are less objective, expensive, time-consuming, and require specialized operators and instrumentation [67]. Overall, the immediate environment of skeletal remains induces specific degradation processes related to the PMI. Micro-CT appears to objectify the results of these degradation processes. Further research into particular environments with well-defined PMIs will be necessary to improve micro-CT’s accuracy further. Comparing these data with data from vibrational spectroscopic analyses of bones, taxonomic identification, preservation mechanisms, diagenetic and thermal alteration pathways, and chemical composition [68] will improve our understating of PMI, especially during the first days.

Comparing these data with data from vibrational spectroscopic analyses of bones, taxonomic identification, preservation mechanisms, diagenetic and thermal alteration pathways, and chemical composition will further improve the estimation of the PMI. This progress will be especially important to help authorities estimate if the found bone is of forensic interest or not. In summary, the following points should be addressed for future research:Further development of prospective technical and scientific protocols.Creating a more extensive data pool of bones with different PMIs, stages of autolysis and putrefaction, where temperature, moisture, insects, depth of burial, and scavenging should be essential factors.

## 5. Conclusions

This study aimed to evaluate the suitability of micro-CT as a non-destructive method for distinguishing between forensic bones with different PMIs as well as archaeological bone material. The main focus of this study was the training of an ANN that allows for estimating the PMI of human bones with micro-CT. The ANN classification results clearly distinguish between forensic and archaeological samples, but also between forensic bone samples with different PMIs. The results show a perfect classification for the fifth class (archaeological samples) and the lowest accuracy for the first class. Micro-CT appears to objectify the results of these degradation processes, and mineral density findings can be used for PMI estimation.

## Figures and Tables

**Figure 1 biology-11-01105-f001:**
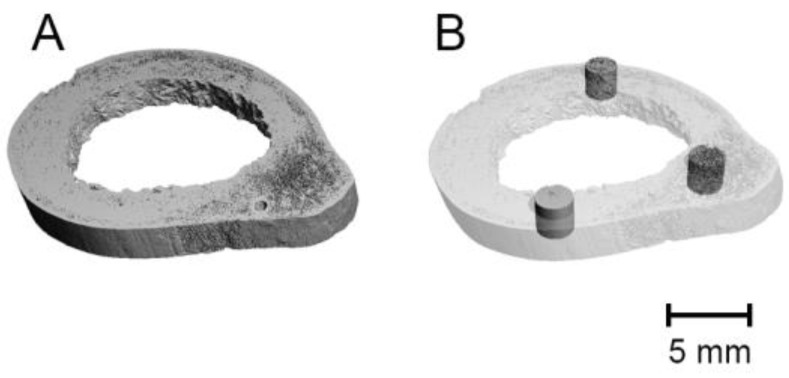
Segmentation and quantitative analysis of the whole cortical bone (**A**) and 3 cylinders aligned centrally in the cortical bone around the analyzed sample (**B**).

**Figure 2 biology-11-01105-f002:**
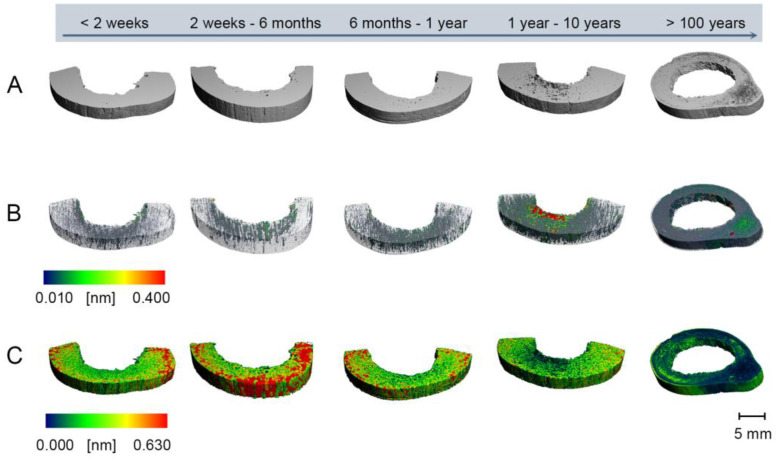
Reconstructions of bone microstructure from micro-CT using the whole cortical bone. (**A**) 3-D surface renders (**B**) Cortical pores (**C**) Connected Cortical Bone Size.

**Figure 3 biology-11-01105-f003:**
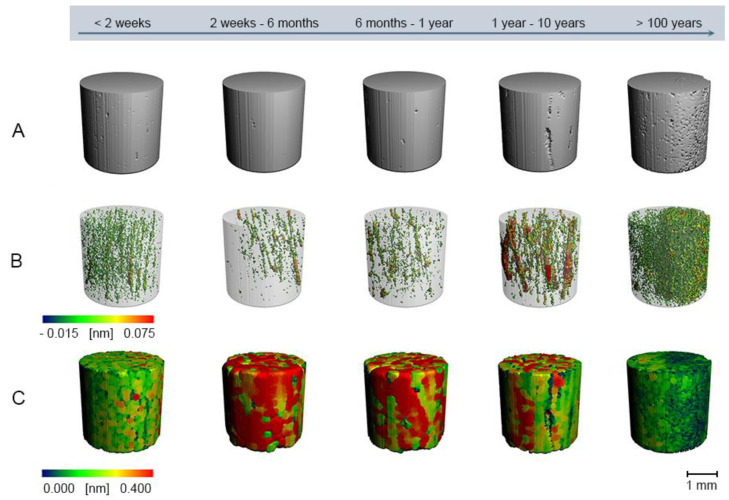
Reconstructions of bone microstructure from micro-CT using cylinders aligned centrally in the cortical bone. (**A**) 3-D surface renders (**B**) Cortical pores (**C**) Connected Cortical Bone Size.

**Figure 4 biology-11-01105-f004:**
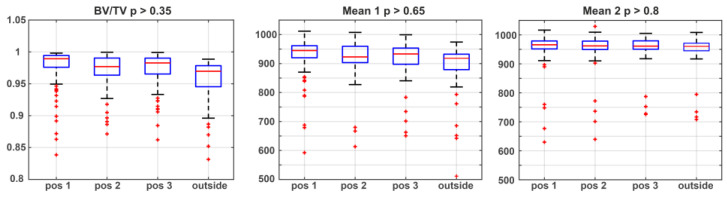
Boxplots of selected values of micro-CT images from all the extraction locations. The Boxplots and the statistical analysis show no significant deviation between a sample taken from the internal of the bones or the external, this result holds for all investigated parameters.

**Figure 5 biology-11-01105-f005:**
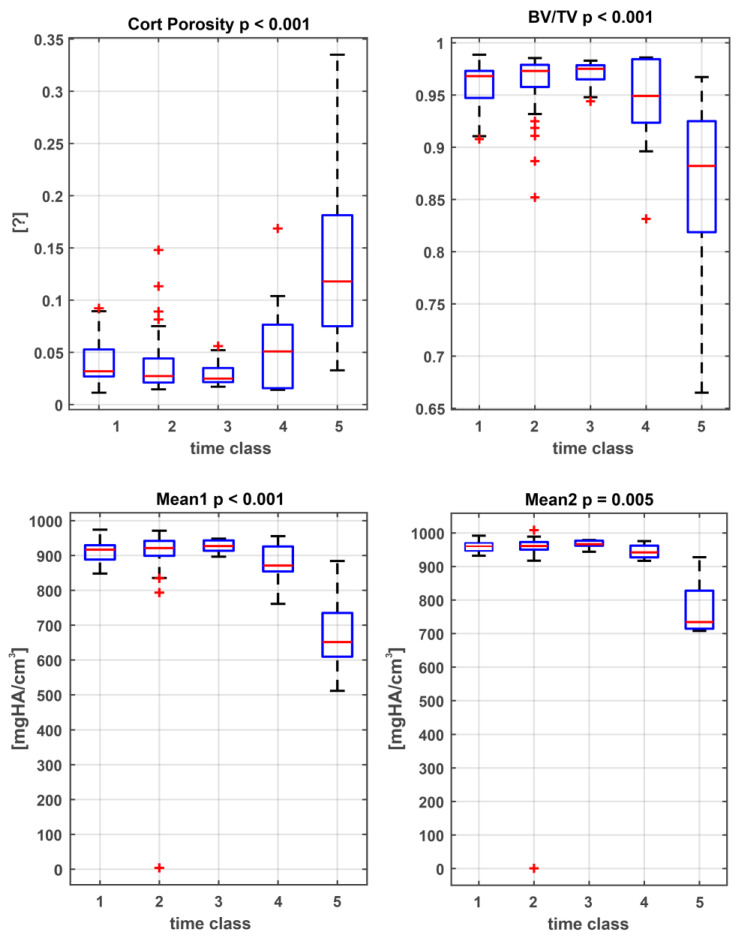
Boxplots of the measured parameters. All parameters show statistically significant differences between the PMI classes (class 1: 0–2 weeks PMI; class 2: 2 weeks–6 months PMI; class 3: 6 months–1 year PMI; class 4: 1 year–10 years PMI; class 5: >100 years PMI).

**Figure 6 biology-11-01105-f006:**
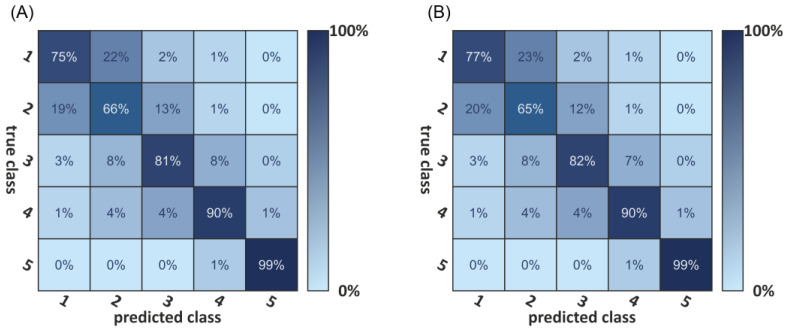
(**A**) Confusion matrix for the neural network-based bone-age classification. (**B**) shows the confusion matrix for the age-adjusted bone-age classification. Classes from 1 to 5 indicate time frames (class 1: 0–2 weeks PMI; class 2: 2 weeks–6 months PMI; class 3: 6 months–1 year PMI; class 4: 1 year–10 years PMI; class 5: >100 years PMI). For both settings, a sample classified to class 5 has a 99% likelihood to belong to this class, that equates to a 1% likelihood for it to be classified to the wrong category.

**Table 1 biology-11-01105-t001:** Micro-CT parameters with associated abbreviations, description, and unit.

Metric Measures	Abbreviation	Description	Standard Unit
Bone volume ratio	BV/TV	Ratio of bone volume to total volume in the ROI	%
Cortical Porosity	Cort Porosity	cortical volume	%
Trabecular number	Tb.N	Mean number of trabeculae per unit length	mm^−1^
Trabecular thickness	Tb.Th	Mean thickness of the trabeculae	mm
Trabecular separation	Tb.Sp	Mean distance between trabeculae	Mm
Apparent density	Mean1	Mean density of the ROI	mgHA/mm^3^
Material density	Mean2	Mean density of the bone fraction of the ROI	mgHA/mm^3^

**Table 2 biology-11-01105-t002:** Parameters are shown for the five different age classes: class 1 with PMI of 0–2 weeks, class 2 with PMI of 2 weeks–6 months, class 3 with PMI of 6 months–1 year, class 4 with PMI of 1 to 10 years, and class 5 with PMI of >100 years. The values with noticeable differences from one representative sample are presented for each class.

Age Class	PMI	Analyzed Areas	BV/TV [%]	Cort Porosity	Tb.N [mm]	Tb.Th [mm]	Tb.Sp [mm]	Mean1 [mgHA/cm^3^]	Mean2 [mgHA/cm^3^]
1	0–2 wk	whole cortical bone	0.96 ± 0.10	0.041 ± 0.022	1.8 ± 0.2	1.40 ± 0.65	2.5 ± 1.2	928 ± 39	950± 121
1	0–2 wk	3 x cylinders aligned centrally	0.95 ± 0.02	n.a	3.1 ± 1.0	0.65 ± 0.22	2.2 ± 1.4	909 ± 31	959 ± 15
2	2 wk–6 mth.	whole cortical bone	0.97 ± 0.027	0.038 ± 0.028	1.8 ± 0.3	1.54 ± 0.60	2.7 ± 1.2	934 ± 41	967 ± 22
2	2 wk–6 mth.	3 x cylinders aligned centrally	0.96 ± 0.02	n.a	3.6 ± 1.4	0.75 ± 0.21	2.9 ± 1.9	894 ± 139	940 ± 142
3	6 mth.–1 yr.	whole cortical bone	0.97 ± 0.02	0.030 ±0.013	1.6 ± 0.2	1.48 ± 0.47	2.5 ± 0.9	939 ± 31	972 ± 12
3	6 mth.–1 yr.	3 x cylinders aligned centrally	0.97 ± 0.01	n.a	2.9 ± 1.2	0.71 ± 0.20	2.3 ± 1.5	926 ± 18	966 ± 11
4	1 yr.–10 yr.	whole cortical bone	0.97 ± 0.02	0.059 ± 0.049	1.8 ± 0.2	1.40 ± 0.78	2.6 ± 1.6	923 ± 37	956 ± 24
4	1 yr.–10 yr.	3 x cylinders aligned centrally	0.94 ± 0.05	n.a	3.6 ± 1.8	0.76 ± 0.20	3.0 ± 2.4	877 ± 59	944 ± 19
5	>100 yr.	whole cortical bone	0.84± 0.23	0.141 ± 0.115	1.4 ± 0.2	0.52 ± 0.40	0.76 ± 0.65	663 ± 162	758 ± 89
5	>100 yr.	3 x cylinders aligned centrally	0.85 ± 0.11	n.a	n.a.	n.a.	n.a.	674 ± 134	776 ± 91

BV/TV [%], ratio between the total analyzed area and the bone compartment of a predefined ROI (region of interest). Cort Porosity, Cortical Porosity, cortical volume in %. Tb.N [1/cm], trabecular number, calculated by taking the inverse of the mean distance between the central axes of the structure [60]. Tb.Th [mm], mean trabecular thickness [66]. Tb.Sp [mm], mean trabecular separation [66]. Mean1 [mg HA/cm^3^], apparent density = Mean density of the bone fraction of the ROI. Mean2 [mg HA/cm^3^], material density = mean density over bone compartment of the region an-alyzed; HA, hydroxylapatite.

## Data Availability

Not applicable.

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
