# Peer review of "Application of Micro-Computed Tomography for the Estimation of the Post-Mortem Interval of Human Skeletal Remains"

_biology, 2022, doi:10.3390/biology11081105_

Round 1

Reviewer 1 Report

Good article. Yet, some improvements could be done: time of death is not the same of PMI. This article tries to estimate PMI ( and not determine). 

The medico legal significance of a case/human reamians is not discussed and it should be. That depends on the legislation of the country in question. It could be 50 years, 20 years, 15 years. That said, the adequacy of the method is challenging. This method should be able to tell whether the remains have, or not, medico legal significance; as it is, it does not. Moreover, and as the authors are well aware, the samples used should be more completed regarding the information on PMI.

Author Response

Dear Editor, Dear Reviewers,

Thank you for considering this paper and for the constructive comments. We changed the paper regarding the comments made by the reviewers. In this response letter, we will document all our answers to the reviewers and state where the changes were applied in the final manuscript: 

Rev1:

Good article. Yet, some improvements could be done: time of death is not the same of PMI. This article tries to estimate PMI ( and not determine).

AW: We deleted time of death and changed from determine to estimate.

The medico legal significance of a case/human reamians is not discussed and it should be. That depends on the legislation of the country in question. It could be 50 years, 20 years, 15 years. That said, the adequacy of the method is challenging. This method should be able to tell whether the remains have, or not, medico legal significance; as it is, it does not. Moreover, and as the authors are well aware, the samples used should be more completed regarding the information on PMI.

AW: We added sentences to discuss the medico legal significance (line 252 and line 297). 

With thanks and kind regards

Johannes Pallua, corresponding author

Priv.-Doz. MMag.Dr.rer.nat. Johannes Pallua MSc PhD

Univ.-Klinik für Orthopädie und Traumatologie

Anichstraße 35

A-6020 Innsbruck

Tel.: +43 50 504 80242

Mail: johannes.pallua@tirol-kliniken.at

Reviewer 2 Report

The authors submitted a manuscript investigating a topic of interest in forensic science, namely PMI estimation. In particular, cortical bone of both forensic and archeological interest was investigated through a microCT approach combined with machine learning.

Although the manuscript is certainly interesting it presents several issues that need to be addressed in order to suitable for publication.

Overall:

Several sentences in the manuscript are often redundant and need to be simplified.

Referencing is exceedingly aged. See point per point below. 

Images definition is poor and often they cannot be interpreted.

Intro:

Line 52-53: the authors correctly introduce the issue of PMI estimation as a challenge in forensic medicine citing two papers dealing with specific approaches on teeth and not a global review on various methods and matrices such as [10.1007/s12024-016-9776-y].

Line 57-58: the authors stated that proposed approaches to estimate PMI lack in reliability and accuracy, corroborating it through citing a 2016 paper in Chinese language. This is not scrupulous and must be addressed. Although on animal model and on a limited PMI window, Locci et al. obtained a rigorous and robust metabolomic approach to estimate PMI, validating it with an independent test set with an error in prediction on 99 minutes on the entire range investigated [10.1007/s00414-020-02468-w]. Most recent literature should be appropriately cited when dealing with such an interesting research field.

In the following lines the authors decide to cite several approaches used to investigated PMI. Metabolomics (see above) and the following papers should be included:

10.1007/s00414-021-02698-6
10.1007/s00414-019-02125-x
10.1038/s41467-021-26318-4

Line 84-87: redundant.

Methods:

Line 96: the fact that the estimation was based on bone whose PMI was judged by eye is a major drawback of the study and should better addressed in the limitation paragraph. The reviewer understand the issue of knowing PMIs in the 'archeological' window but forensic ones must be known in order to obtain a rigorous model. Anyway the presumptive range of PMI should be furnished.

Line 98-100: if data are not shown, and such contribution would not add anything to the scope of the study, why do the authors need to include this sentence in the text?

Line 128: might be a typo.

Line 159-161: on which bases was it normalized to 40 years old and not to a older age?

Results: are very difficult to interpret as the images definition is poor.

Discussion:

The section barely discuss results and should be implemented.

Line 250-251: is there really a need for new tools to distinguish a few weeks bone from a thousand years one?

How do the authors explain first and second classes lower predictive ability of their system with respect to the fact the worked on unknown PMIs?(as they stated in methods)

To the reader it is not clear the error of the proposed approach. This is a mandatory step to asses reproducibility of the methods required by the rigorous forensic criteria.

The non-destructive evaluation is a point of strength of this manuscript and should be better highlighted.

Others:

Author contributions should be certainly simplified. Funding acquisition (JP) collides with the declaration of no funding few lines below.

Line 33-35: the reported 99% in the simple summary might be a bit overwhelming with respect to the full results. Sentences are also redundant.

Line 36-37: redundant.  

Author Response

Dear Editor, Dear Reviewers,

Thank you for considering this paper and for the constructive comments. We changed the paper regarding the comments made by the reviewers. In this response letter, we will document all our answers to the reviewers and state where the changes were applied in the final manuscript:

Rev2:

The authors submitted a manuscript investigating a topic of interest in forensic science, namely PMI estimation. In particular, cortical bone of both forensic and archeological interest was investigated through a microCT approach combined with machine learning.

Although the manuscript is certainly interesting it presents several issues that need to be addressed in order to suitable for publication.

Overall:

Several sentences in the manuscript are often redundant and need to be simplified.

Referencing is exceedingly aged. See point per point below.

Line 52-53: the authors correctly introduce the issue of PMI estimation as a challenge in forensic medicine citing two papers dealing with specific approaches on teeth and not a global review on various methods and matrices such as [10.1007/s12024-016-9776-y].

AW: We added the suggested citation.

Images definition is poor and often they cannot be interpreted.

AW: Image defintions were extended for better reading.

Line 57-58: the authors stated that proposed approaches to estimate PMI lack in reliability and accuracy, corroborating it through citing a 2016 paper in Chinese language. This is not scrupulous and must be addressed. Although on animal model and on a limited PMI window, Locci et al. obtained a rigorous and robust metabolomic approach to estimate PMI, validating it with an independent test set with an error in prediction on 99 minutes on the entire range investigated [10.1007/s00414-020-02468-w]. Most recent literature should be appropriately cited when dealing with such an interesting research field.

AW: We removed the citation 2016 in Chinese language and added the citation of Locci et al.

In the following lines the authors decide to cite several approaches used to investigated PMI. Metabolomics (see above) and the following papers should be included:

10.1007/s00414-021-02698-6

10.1007/s00414-019-02125-x

10.1038/s41467-021-26318-4

AW: We added the recommended literature in the text.

Line 84-87: redundant.

AW: We deleted the redundant information.

Methods:

Line 96: the fact that the estimation was based on bone whose PMI was judged by eye is a major drawback of the study and should better addressed in the limitation paragraph. The reviewer understand the issue of knowing PMIs in the 'archeological' window but forensic ones must be known in order to obtain a rigorous model. Anyway the presumptive range of PMI should be furnished.

AW: The PMI was estimated by police investigations (when a person was seen the last time, date oft he newspaper in front oft he door, the last phone call or e-mail…). Our goal was to show differences in the bones structure between forensic and archaelogical bones. For furture research and to obatin a rigorous model, bones with an exact PMI and the same environmental influences should be examined. A study like this takes much more time, but will be a future interest.

Line 98-100: if data are not shown, and such contribution would not add anything to the scope of the study, why do the authors need to include this sentence in the text?

AW: We deleted the missleading information.

Line 128: might be a typo.

AW: We apologize, but we did not find the typo.

Line 159-161: on which bases was it normalized to 40 years old and not to a older age?

AW: There was no distinct reason to pick 40 years, or any other age for that matter. Since all the samples were normalized to the same age it does not make a difference which age is picked. We added the following information: The data of [59] demonstrated a linear or quadratic behaviour between the ages of 20 to 60. That made it numerically favourable to choose an age in this range. From this range, the median age was taken. However, the choice for the base age is not expected to change the outcome of the analysis since all data were normalized to that age. Therefore, the values have been normalized to the 40-year age value, and data compared between the different age classes as described above. This has been clarified in the methods section.

Results: are very difficult to interpret as the images definition is poor.

AW: We imporved the image quality and definitions.

Discussion:

The section barely discuss results and should be implemented.

AW: We added sentences regarding the results in the discussion part (line 253..).

Line 250-251: is there really a need for new tools to distinguish a few weeks bone from a thousand years one?

AW: We explained the aim of the study more precisely. We corrected the sentence also in the conclusion part.

How do the authors explain first and second classes lower predictive ability of their system with respect to the fact the worked on unknown PMIs?(as they stated in methods)

AW: For clarification, this issue is discussed in lines 297 ff.

To the reader it is not clear the error of the proposed approach. This is a mandatory step to asses reproducibility of the methods required by the rigorous forensic criteria.

AW: The confusion matrices in figure 6 describe the accuracy of the method, i.e. the true positive rate of the neural network. Considering a sample classified in class one for instance. This sample has a 75% likelihood to originate in class 1, a 32% likelihood to originate from class 2, a 2% likelihood to belong to class 3 and so on. A sample classified to class 5 has a 99% likelihood to belong to this class, that equates to a 1% likelihood for it to be classified to the wrong category. This has been added to the figure legend.

The non-destructive evaluation is a point of strength of this manuscript and should be better highlighted.

AW: We added „non-destructive“ in the manuscript in the discussion and the conclusions. And that the method is non-invasive is also mentioned in line 277 in the discussion. The advantages of the method compared to other methods are also presented in line 278.

Others:

Author contributions should be certainly simplified. Funding acquisition (JP) collides with the declaration of no funding few lines below.

AW: We deleted the misleading information.

Line 33-35: the reported 99% in the simple summary might be a bit overwhelming with respect to the full results. Sentences are also redundant.

AW: The sentences were adapted.

Line 36-37: redundant.

AW: We deleted the redundant information.

With thanks and kind regards

Johannes Pallua, corresponding author

Priv.-Doz. MMag.Dr.rer.nat. Johannes Pallua MSc PhD

Univ.-Klinik für Orthopädie und Traumatologie

Anichstraße 35

A-6020 Innsbruck

Tel.: +43 50 504 80242

Mail: johannes.pallua@tirol-kliniken.at

Round 2

Reviewer 2 Report

The authors accept to revise the manuscript, which is now improved.

Some minor issues still remain.

Ref. [6] added does not fit with '...serious drawbacks in terms of reliability and accuracy'. Might be moved later among methods used to estimate PMIs in esperimentai settings as 'metabolomics'.

Line 96 previous comment should be better addressed in limitations.

Author contributions redundant and should be simplified.

Author Response

Dear Editor, Dear Reviewers,

Thank you for considering this paper and for the constructive comments. We changed the paper regarding the comments made by the reviewer 2. In this response letter, we will document all our answers to the reviewer and state where the changes were applied in the final manuscript:

Rev2

Some minor issues still remain.

Ref. [6] added does not fit with '...serious drawbacks in terms of reliability and accuracy'. Might be moved later among methods used to estimate PMIs in esperimentai settings as 'metabolomics'.

AW: We removed Ref. [6] and added metabolomics with Ref. [6] to the suggested position.

Line 96 previous comment should be better addressed in limitations.

AW: We added the following sentences in the discussion limitation section: PMI was classified based on police investigations and forensic needs. The average PMI result was used for classification when conventional estimation was uncertain. Indicated in yellow.

Author contributions redundant and should be simplified.

AW: We simplified author contributions.

With thanks and kind regards

Johannes Pallua, corresponding author